# Mixture of Groups: Grouped Gating and Cross Mixing for Parameter-Efficient LLM Fine-Tuning

## Abstract

Low-rank adaptation (LoRA) is a widely used parameter-efficient fine-tuning (PEFT) method for large language models. However, by learning independent adapters for each layer, LoRA and its variants ignore the inherent functional similarity between adjacent layers, limiting their potential to fully exploit the hierarchical representations across depth. To address this, we propose Mixture of Groups (MoG), a novel group-sharing framework that partitions layers into functional groups, shares low-rank adapters within each group, and employs adaptive gating and cross-mixing mechanisms to enable flexible fine-tuning. This approach leverages inter-layer similarity to capture both commonalities and unique characteristics across layers, achieving a more efficient and expressive subspace than LoRA. Moreover, MoG is designed as a plug-and-play framework that can be seamlessly integrated into other PEFT methods such as DoRA and PiSSA to boost their performance. Extensive experiments on multiple benchmarks demonstrate that MoG achieves overall superior performance compared with prior methods under comparable parameter budgets, highlighting its ability to combine efficiency with strong downstream effectiveness.

## 1 Introduction

The rapid scaling of large language models (LLMs) (Achiam et al., 2023; Dubey et al., 2024; Guo et al., 2025) has brought remarkable advances in natural language processing, showcasing strong language understanding and generation capabilities. However, the massive parameter counts incur prohibitive computational and storage costs, making full-model fine-tuning impractical in resource-constrained settings. To address this, parameter-efficient fine-tuning (PEFT) (Houlsby et al., 2019; Lester et al., 2021; Liu et al., 2022) has emerged, aiming to adapt LLMs by training only a small set of additional parameters. Among various PEFT methods, LoRA (Hu et al., 2022) freezes pre-trained weights and inserts trainable low-rank matrices, substantially reducing trainable parameters while often preserving or even surpassing full fine-tuning performance.

Despite its success, LoRA trains independent adapters for each layer, leading to redundancy. To mitigate this, several sharing-based methods have been proposed. VeRA (Kopiczko et al., 2024) shares a global pair of low-rank matrices across all layers and uses layer-specific scaling vectors. VB-LoRA (Li et al., 2024) builds a global vector library with top-$k$ mixing, while RaSA (He et al., 2025) partially shares ranks across layers to expand expressiveness at constant parameter budgets. In another approach, DenseLoRA (Mu et al., 2025) employs global encoder-decoder sharing while learning lightweight dense matrices per layer. These strategies achieve high compression but face a key limitation: they rely heavily on global sharing, which overlooks the inherent functional diversity across layers, thereby limiting the abilities of different layers to develop specialized adaptation strategies and ultimately hampering expressiveness and task performance.

Recent studies reveal that adjacent layers in LLMs often exhibit functional similarity. Jiang et al. (2025) showed that adjacent layers yield highly correlated representations, with similarity decaying as layer distance increases. Min & Wang (2025) demonstrated that adjacent weights are often clustered, indicating functional regions along model depth. Our analysis using Distributed Cosine Similarity (DOCS) (Min & Wang, 2025) further confirms this trend. As shown in Figure 1(a),

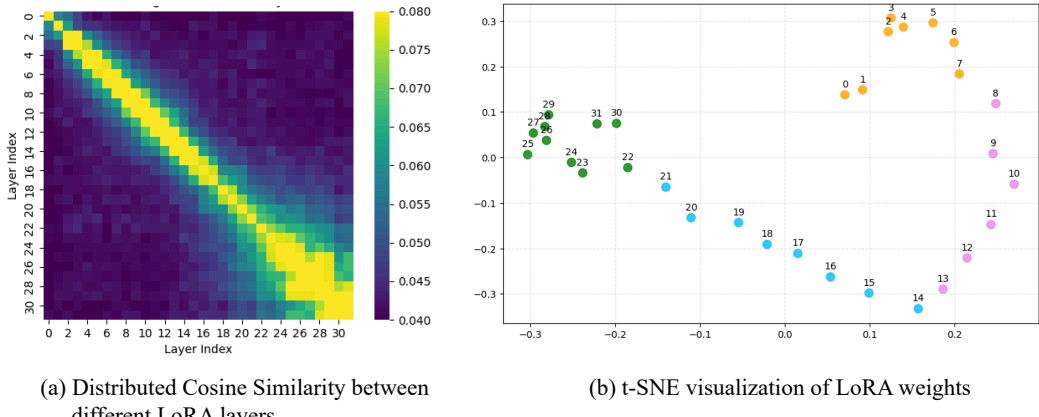

(a) Distributed Cosine Similarity between
different LoRA layers

(b) t-SNE visualization of LoRA weights

Figure 1: (a) Heatmap of distributed cosine similarity between different LoRA layers on LLaMA2-7B. (b) t-SNE visualization of LoRA weights clustered by layer indices on LLaMA2-7B.

LoRA weights exhibit bright diagonal bands in the similarity heatmap, reflecting strong correlations between neighboring layers. Figure 1(b) further illustrates this pattern: adjacent layers often cluster together along depth, suggesting similar functional subspaces. Importantly, these clusters are not completely isolated from each other, and even within the same cluster, individual layers retain subtle distinctions. Such observations motivate three design principles underlying our method: group sharing to reduce redundancy among similar layers, adaptive gating to flexibly combine information across groups, and per-layer fusion matrices to capture the unique characteristics of each layer.

Motivated by this, we propose **Mixture of Groups (MoG)**, a group-sharing PEFT framework that leverages inter-layer commonality to reduce redundancy while maintaining the capacity for layer-specific specialization. Our method partitions layers into groups with shared low-rank adapters, employs a lightweight per-layer gating network to adaptively aggregate group parameters, and introduces a cross-mixing module that further enhances layer-wise specialization. Furthermore, we demonstrate that MoG is a versatile framework that generalizes to other PEFT methods such as DoRA (Liu et al., 2024) and PiSSA (Meng et al., 2024), highlighting its broad applicability.

In summary, our contributions are as follows:

- We introduce MoG, a novel PEFT framework that balances parameter sharing and layer-specific adaptation by combining group-wise sharing, adaptive gated aggregation, and cross-mixing, achieving efficient and expressive fine-tuning.

- We demonstrate MoG's generality: it integrates seamlessly with LoRA and extends naturally to other PEFT methods such as DoRA and PiSSA, offering a unified design paradigm.

- We validate MoG through extensive experiments and analyses on multiple LLMs, showing consistent efficiency gains and strong performance under diverse parameter budgets.

## 2 RELATED WORK

**Low-Rank Adaptation (LoRA).** As one of the most widely used PEFT methods, LoRA (Hu et al., 2022) freezes pre-trained weights and injects trainable low-rank matrices, effectively reducing trainable parameters while maintaining performance. Building on LoRA, researchers have proposed various improved methods. AdaLoRA(Zhang et al., 2023) uses singular value decomposition to adaptively allocate ranks based on the importance scores of the weight matrix. DoRA(Liu et al., 2024) decomposes the pre-trained weights into amplitude and direction components for fine-tuning. PiSSA (Meng et al., 2024) initializes low-rank matrices with principal components and updates them during training for faster convergence. FLoRA (Si et al., 2024) generalizes LoRA to higher-order tensors using Tucker decomposition. HiRA(Huang et al., 2025) enhances the model's expressive power by retaining higher-order update parameters using the Hadamard product. These variants highlight LoRA's flexibility but still rely on independent adapters per layer.

**Parameter Sharing in LoRA.** To further reduce redundancy, various parameter-sharing approaches have been proposed. VeRA (Kopiczko et al., 2024) shares a global random matrix and learns layer-specific scaling vectors. Tied-LoRA (Renduchintala et al., 2024) employs weight tying and selective updates. VB-LoRA(Li et al., 2024) proposes a "divide-and-share" mechanism based on a global vector library and top-k hybrid modules across modules and layers. To capture both local and global information, BSLoRA (Zhou et al., 2025) combines intra-layer and inter-layer sharing, MoS(Wang et al., 2025) constructs low-rank adaptation matrices via flexible combinations of shards, and Lily(Zhong et al., 2025) introduces an interconnected expert framework. Some designs aim to balance compression and expressiveness. RaSA (He et al., 2025) shares partial ranks across layers to expand effective capacity, while DenseLoRA(Mu et al., 2025) shares an encoder–decoder and adds small dense matrices per layer. Although these methods reduce overhead, their reliance on global sharing overlooks functional and structural diversity across layers. In contrast, our approach exploits adjacent-layer similarity to form functional groups, where localized sharing and adaptive mixing achieve efficient parameterization while retaining model expressiveness.

## 3 COMPARATIVE SHARING STRATEGIES

LoRA adapts a low-rank decomposition update weight $\Delta W$ while keeping the pre-trained weight $W_0 \in \mathbb{R}^{d \times k}$ frozen. The fine-tuned weight $W$ can be represented as:

$$W = W_0 + \Delta W = W_0 + AB, \tag{1}$$

where $A \in \mathbb{R}^{d \times r}$ and $B \in \mathbb{R}^{r \times k}$, with rank $r \ll min(d, k)$. While effective, LoRA assigns an independent $(A, B)$ pair to every layer, introducing redundancy and storage overhead.

**Global Sharing.** Global sharing uses a single pair of low-rank adapters $(A, B)$ across all $L$ layers. This design achieves strong compression compared with LoRA. When both have identical parameter budgets, global sharing concentrates the entire budget into a single subspace with an effective rank of $Lr$, thus maximizing overall capacity in terms of rank. However, enforcing all layers to operate in the same low-rank subspace suppresses layer-specific adaptation and weakens the hierarchical representation ability of the model, often limiting downstream performance.

**Group Sharing.** To balance redundancy reduction and layer diversity, we test a simple group sharing strategy: dividing $L$ layers into $n$ groups, each group sharing a pair $(A_g, B_g)$, $g = 1, \ldots, n$. If layer $i$ belongs to group $g$, its update is simply:

$$\Delta W_i = A_g B_g. \tag{2}$$

This method utilizes the functional similarity of adjacent layers to reduce redundancy, while allowing different groups to maintain distinct functionality. We conduct experiments on DeBERTa-v3(He et al., 2023) using the GLUE(Wang et al., 2018) benchmark, with results shown in Table 1.

Table 1: Results on the GLUE subset with DeBERTa-v3. The notation $r \times n$ indicates adapter rank $r$ and the number of groups $n$. All methods are compared under the same parameter budget.

| Method | $r \times n$ | CoLA | RTE | MRPC | STS-B | Avg. |
|---|---|---|---|---|---|---|
| LoRA | $8 \times 12$ | 69.82 | 85.20 | 89.95 | 91.60 | 84.14 |
| Global Sharing | $96 \times 1$ | 70.17 | 86.28 | 90.44 | 90.53 | 84.36 |
| Group Sharing | $16 \times 6$ | 71.26 | 87.00 | 90.93 | 90.82 | 85.00 |
| | $24 \times 4$ | 71.18 | **87.36** | 91.40 | 91.20 | 85.29 |
| | $32 \times 3$ | **71.59** | 86.64 | **91.42** | **91.64** | **85.32** |

Under the same parameter budget, global sharing improves only marginally over LoRA, indicating that concentrating capacity into a single shared subspace fails to capture layer-specific variations. In contrast, group sharing consistently achieves higher average accuracy across different configurations. It strikes a favorable balance: compared to LoRA, it reduces unnecessary redundancy; compared to global sharing, it maintains the crucial hierarchical capacity, leading to superior performance. However, the simple group sharing still lacks flexibility in capturing differences across groups. To address this, we introduce MoG in the next section, which extends group sharing with adaptive mechanisms for greater flexibility.

# 4 METHOD

## 4.1 MIXTURE OF GROUPS

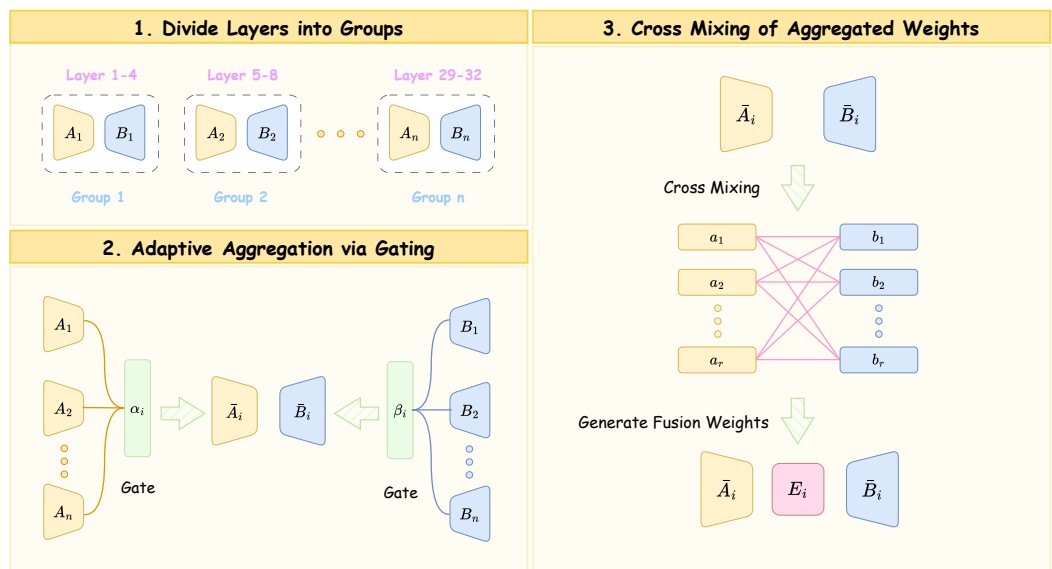

Figure 2: Overview of MoG framework. (1) Layers are divided into groups, each with shared low-rank adapters $(A_g, B_g)$. (2) Gating networks generate layer-specific coefficients to aggregate group adapters into $(\bar{A}_i, \bar{B}_i)$. (3) A cross-mixing matrix $E_i$ fuses the aggregated weights, yielding the final update $\Delta W_i = \bar{A}_i E_i \bar{B}_i$.

To reduce inter-layer parameter redundancy while maintaining intra-layer modeling capabilities, we propose the MoG framework. It consists of three components: (1) grouped sharing to reduce redundancy, (2) aggregation via gating to allow layers to flexibly combine shared parameters, and (3) cross mixing to enrich the representational capacity. The overall workflow is shown in Figure 2.

**Grouped Sharing.** Instead of learning independent adapters for every layer in LoRA, we partition the $L$ transformer layers into $n$ groups, each containing several consecutive layers, and associated with a pair of low-rank matrices $(A_g, B_g)$:

$$\mathcal{A} = \{A_1, A_2, \ldots, A_n\},$$
$$\mathcal{B} = \{B_1, B_2, \ldots, B_n\}, \tag{3}$$

where $A_g \in \mathbb{R}^{d \times r}$, $B_g \in \mathbb{R}^{r \times k}$. In this way, each layer selects and combines parameters from the global pools $\mathcal{A}, \mathcal{B}$, substantially reducing parameter redundancy.

**Adaptive Aggregation via Gating.** To make use of shared parameters flexibly, each layer $i$ maintains learnable gating scores $s_i, t_i \in \mathbb{R}^n$. Normalized by softmax, they produce gating coefficients $\alpha_i = \text{softmax}(s_i)$ and $\beta_i = \text{softmax}(t_i)$. These coefficients are then used to form layer-specific aggregated matrices, obtained as weighted combinations of group parameters:

$$\bar{A}_i = \sum_{g=1}^{n} \alpha_{ig} A_g, \;\; \bar{B}_i = \sum_{g=1}^{n} \beta_{ig} B_g, \tag{4}$$

where $\bar{A}_i$ and $\bar{B}_i$ denote the aggregated adapters for layer $i$. This mechanism enables each layer to adaptively combine information from multiple groups, rather than relying on a single shared adapter.

**Cross Mixing of Aggregated Weights.** Although $(\bar{A}_i, \bar{B}_i)$ already introduces flexibility, the differentiated capabilities of each layer remain constrained. To enrich the representation capacity of each layer, we further decompose $\bar{A}_i$ and $\bar{B}_i$ into vector sets:

$$\bar{A}_i = [a_1, a_2, \ldots, a_r], \quad \bar{B}_i = [b_1, b_2, \ldots, b_r], \tag{5}$$

where $a_p \in \mathbb{R}^{d \times 1}$, $b_q \in \mathbb{R}^{1 \times k}$. For each pair $(a_p, b_q)$, we introduce a learnable fusion weight $e_{pq}$, and form the update by cross mixing:

$$\Delta W_i = \sum_{p=1}^{r} \sum_{q=1}^{r} e_{pq} a_p b_q. \tag{6}$$

Equivalently, it can be written as:

$$\Delta W_i = \bar{A}_i E_i \bar{B}_i, \tag{7}$$

where $E_i = [e_{pq}] \in \mathbb{R}^{r \times r}$ is the cross-mixing weight. This cross-vector mixing mechanism enables each layer to not only utilize shared information within the group, but also capture cross-group interactions through $E_i$, thereby enhancing the model's expressive power.

## 4.2 PARAMETER ANALYSIS AND INITIALIZATION

**Parameter Complexity.** Let the input dimension be $d$, the output dimension $k$, the number of layers $L$, the number of groups $n$, and the adapter rank $r$. For LoRA, the trainable parameters are $P_{\text{LoRA}} = Lr(d + k)$. For MoG, they are $P_{\text{MoG}} = nr(d + k) + Lr^2 + Ln$, consisting of group sharing, fusion, and gating terms. Since $r \ll \min(d, k)$, the fusion term $Lr^2$ and the gating term $Ln$ are negligible compared with $nr(d + k)$. Therefore, as long as $n < L$, MoG achieves effective compression relative to standard LoRA, while retaining additional flexibility. A comparison of parameter counts is summarized in Table 2.

Table 2: Parameter counts of LoRA and MoG. A concrete example with $L$=32, $d$=$k$=1024, $r$=8, and $n$=4 is provided in Appendix G.

| Method | Parameter Formula | Example ($L$=32, $d$=1024, $r$=8) |
|--------|-------------------|-----------------------------------|
| LoRA | $Lr(d + k)$ | 524,288 |
| MoG | $nr(d + k) + Lr^2 + Ln$ | 67,712 |

**Initialization Strategy.** Matrix $A$ is initialized using the Kaiming initialization(He et al., 2015), and matrix $B$ is initialized to zero, ensuring that $\Delta W_i$ is zero at the beginning. $E_i$ is initialized with the Kaiming initialization, following the findings of MoSLoRA (Wu et al., 2024) that such initialization improves stability and performance when mixing subspaces. The gate of each layer is initialized with the one-hot initialization, 1 for its assigned group and 0 otherwise. It provides a stable initialization where each layer starts with a clear preference for one group. During training, the gates gradually adapt into soft distributions, enabling effective cross-group interactions.

## 4.3 ADVANTAGE ANALYSIS

**Architectural Flexibility.** MoG provides a highly flexible architecture. Specifically, when the number of groups equals the number of layers ($n = L$), and the fusion matrix $E$ reduces to the identity, MoG is equivalent to standard LoRA. When $n = 1$, all layers share the same $(A, B)$ pair, and MoG degenerates into the structure of DenseLoRA: hidden states are first projected by $A$, transformed by the small dense matrix $E$, and then reconstructed by $B$. Thus, MoG subsumes both independent-adapter and fully-shared settings within a single framework.

**Universality as a Fine-Tuning Paradigm.** The core of MoG is to build a unified parameter-efficient fine-tuning paradigm through group sharing and cross-mixing. It is not only limited to LoRA, but can be extended to other PEFT methods naturally.

- **DoRA:** Using Eq. 7, MoG first produces fused updates $\Delta W$. Then DoRA is applied, fine-tuning the magnitude and direction separately to preserve its design.
- **PiSSA:** Groups are first constructed, and shared $(A, B)$ matrices can be initialized from the average pre-trained weights within each group. Since PiSSA primarily focuses on the initialization stage of LoRA, we do not introduce the mixed weights $E$ but maintain the remainder of the process consistent with PiSSA, which only updates the principal components while freezing the rest.

**Expressive Power.** MoG enlarges the representational capacity compared to LoRA. The updated matrix can be represented as a weighted combination of $n^2$ rank $r$ matrices, resulting in the space of MoG admitting at most $n^2 r$ effective degrees of freedom, while standard LoRA is limited to $r$. Consequently, the subspace of LoRA is contained in that of MoG: $\mathcal{S}_{\text{LoRA}} \subseteq \mathcal{S}_{\text{MoG}}$. This implies that MoG retains the capacity of LoRA while providing greater expressiveness. The detailed proof is provided in Appendix C.

## 5 EXPERIMENTS

### 5.1 SETTINGS

**Datasets.** We conduct experiments on four different benchmarks. For **commonsense reasoning**, we use a dataset comprising 170k training samples(Hu et al., 2023) that includes BoolQ(Clark et al., 2019), PIQA(Bisk et al., 2020), SIQA(Sap et al., 2019), HellaSwag(Zellers et al., 2019), WinoGrande(Sakaguchi et al., 2021), ARC-easy and ARC-challenge(Clark et al., 2018), and Open-bookQA(Mihaylov et al., 2018). For **mathematical reasoning**, we train on MetaMath(Yu et al., 2024) and test on GSM8K(Cobbe et al., 2021) and MATH(Yu et al., 2024). For **code generation**, we train on CodeFeedback(Zheng et al., 2024) and evaluate on HumanEval(Chen et al., 2021) and MBPP(Austin et al., 2021). For **natural language understanding**, we fine-tune on the GLUE benchmark(Wang et al., 2018), which covers a diverse set of classification, entailment, and semantic similarity tasks.

**LLMs.** We fine-tune three representative models: LLaMA2-7B (Touvron et al., 2023), LLaMA3-8B (Dubey et al., 2024), and DeBERTa-V3 (He et al., 2023).

**Baselines.** We compare MoG to LoRA and several parameter-sharing PEFT methods:

- **LoRA** (Hu et al., 2022) freezes the pretrained model weights and injects trainable rank decomposition matrices into each layer of the Transformer architecture.
- **RaSA** (He et al., 2025) enhances the expressive capacity of LoRA by leveraging partial rank sharing across layers.
- **DenseLoRA** (Mu et al., 2025) incorporates a single Encoder-Decoder to refine and compress hidden representations across all adaptation layers before applying adaptation.

In addition, we also apply the MoG architecture to other methods such as **DoRA** (Liu et al., 2024) and **PiSSA** (Meng et al., 2024). Details of the application scheme are provided in Section 4.3.

**Training Details.** We set the LoRA rank to 32 for LLaMA2-7B and LLaMA3-8B, while the rank is set to 8 for DeBERTa-V3. For parameter-sharing methods, we adjust the ranks accordingly to ensure that the number of trainable parameters remains comparable across methods. The detailed hyperparameter configurations are provided in the Appendix B.

### 5.2 MAIN RESULTS

Table 3: Performance of LLaMA3-8B on commonsense reasoning datasets. Results for LoRA and DoRA are sourced from Liu et al. (2024). The entries DoRA(+MoG) and PiSSA(+MoG) represent the results of integrating our MoG framework with the DoRA and PiSSA fine-tuning methods.

| Method | # Params (%) | BoolQ | PIQA | SIQA | HellaS | WinoG | ARC-e | ARC-c | OBQA | Avg. |
|--------|--------------|-------|------|------|--------|-------|-------|-------|------|------|
| LoRA | 0.70 | 70.8 | 85.2 | 79.9 | 91.7 | 84.3 | 84.2 | 71.2 | 79.0 | 80.8 |
| DenseLoRA | 0.70 | 74.4 | 88.4 | 80.7 | 95.3 | 86.7 | 91.5 | 82.0 | 85.6 | 85.6 |
| RaSA | 0.70 | 74.7 | 89.4 | 80.6 | 94.5 | 86.4 | 91.3 | 81.7 | 87.0 | 85.7 |
| MoG | 0.70 | 75.6 | 89.5 | 80.0 | 95.4 | 86.5 | 92.1 | 82.7 | 88.2 | **86.3** |
| DoRA | 0.71 | 74.6 | 89.3 | 79.9 | 95.5 | 85.6 | 90.5 | 80.4 | 85.8 | 85.2 |
| DoRA(+MoG) | 0.71 | 74.8 | 89.2 | 80.9 | 95.4 | 86.4 | 91.9 | 82.4 | 86.6 | **86.0** |
| PiSSA | 0.70 | 73.2 | 89.0 | 80.2 | 95.0 | 86.7 | 91.1 | 81.7 | 85.5 | 85.3 |
| PiSSA(+MoG) | 0.70 | 73.3 | 89.7 | 81.2 | 95.5 | 87.8 | 91.8 | 81.2 | 88.0 | **86.1** |

**Results on Commonsense Reasoning.** As shown in Table 3, LoRA reaches an average of 80.8%, while introducing MoG improves it to 86.3%, resulting in a significant gain of 5.5%. Compared with parameter-sharing methods, MoG outperforms DenseLoRA and RaSA by 0.7% and 0.6%, indicating that MoG better balances parameter sharing and differentiated modeling compared to global sharing strategies, thereby constructing a more expressive update space. Moreover, integrating MoG with DoRA and PiSSA further boosts performance, resulting in improvements from 85.2% to 86.0% and from 85.3% to 86.1%, respectively. This further demonstrates that MoG is not limited to LoRA but can serve as a general-purpose enhancement module to effectively improve the performance of different PEFT methods.

**Results on Mathematical Reasoning & Code Generation.** In mathematical reasoning and code generation tasks, MoG consistently surpasses LoRA (Table 4). For LLaMA2-7B, MoG improves GSM8K by 2.6% and MATH by 2.0%, and it outperforms LoRA by 2.4% and 2.9% on code generation tasks. On LLaMA3-8B, MoG further improves its performance, demonstrating stronger generalisation capabilities. Furthermore, we introduce MoG into experiments on DoRA and PiSSA (Table 5). DoRA(+MoG) and PiSSA(+MoG) outperform the baseline methods across all four tasks, further validating its applicability and advantages as a general-purpose module.

Table 4: Performance of fine-tuning methods on LLaMA2-7B and LLaMA3-8B for mathematical reasoning and code generation tasks.

| Model | Method | # Params (%) | GSM8K | MATH | HumanEval | MBPP |
|---|---|---|---|---|---|---|
| LLaMA2-7B | LoRA | 1.17 | 59.2 | 10.4 | 31.7 | 39.2 |
| | RaSA | 1.17 | 60.6 | 11.9 | 35.4 | 41.0 |
| | DenseLoRA | 1.17 | 60.9 | 12.1 | **35.5** | 41.6 |
| | MoG | 1.17 | **61.8** | **12.4** | 34.1 | **42.1** |
| LLaMA3-8B | LoRA | 1.03 | 79.8 | 39.9 | 60.8 | 70.2 |
| | RaSA | 1.03 | 81.6 | 41.1 | 62.2 | 71.9 |
| | DenseLoRA | 1.03 | 81.4 | 40.5 | 62.8 | 72.8 |
| | MoG | 1.03 | **81.8** | **41.6** | **63.4** | **74.7** |

Table 5: Performance of DoRA and PiSSA augmented with MoG on mathematical reasoning and code generation tasks.

| Model | Method | # Params (%) | GSM8K | MATH | HumanEval | MBPP |
|---|---|---|---|---|---|---|
| LLaMA2-7B | DoRA | 1.19 | 59.6 | 10.7 | 31.0 | 40.7 |
| | DoRA(+MoG) | 1.19 | **60.1** | **11.5** | **32.3** | **43.5** |
| | PiSSA | 1.17 | 59.6 | 11.1 | 31.1 | 40.3 |
| | PiSSA(+MoG) | 1.17 | **60.5** | **11.7** | **35.0** | **41.5** |
| LLaMA3-8B | DoRA | 1.05 | 81.4 | 40.7 | 63.4 | 73.0 |
| | DoRA(+MoG) | 1.05 | **82.4** | **41.0** | **65.2** | **73.8** |
| | PiSSA | 1.03 | 81.0 | 40.0 | 62.6 | 70.8 |
| | PiSSA(+MoG) | 1.03 | **81.9** | **40.3** | **64.6** | **72.8** |

**Results on the GLUE Benchmark.** Table 6 summarizes results on DeBERTa-V3. MoG achieves 89.46% average, 1.1% higher than LoRA and better than DenseLoRA and VBLoRA. When combined with DoRA and PiSSA, MoG still demonstrates gains. DoRA(+MoG) reaches 89.46% and PiSSA(+MoG) 88.94%, both exceeding their originals. The experimental results validate its broad applicability and robustness in natural language understanding tasks.

## 5.3 ABLATION STUDY

To assess the contribution of MoG's key components, we perform ablations on the commonsense reasoning benchmark, with results shown in Table 10. The complete MoG method achieves an average performance of 86.3%, outperforming variants with removed components. Eliminating the cross-mixing matrix $E_i$ reduces performance to 85.1% (–1.2%), confirming that mixing weights

Table 6: Results on the GLUE benchmark with the DeBERTa-V3 base model. The results of LoRA are sourced from Hu et al. (2022).

| Method | # Params (%) | MNLI | SST-2 | CoLA | QQP | QNLI | RTE | MRPC | STS-B | Avg. |
|---|---|---|---|---|---|---|---|---|---|---|
| LoRA | 0.71 | 90.65 | 94.95 | 69.82 | **91.99** | 93.87 | 85.20 | 89.95 | 91.60 | 88.34 |
| VBLoRA | 0.71 | 88.66 | 95.41 | 68.41 | 88.83 | 92.73 | 84.48 | 90.20 | 91.29 | 87.50 |
| DenseLoRA | 0.71 | 90.49 | **96.44** | 70.49 | 91.10 | **94.56** | 87.73 | 90.93 | 91.77 | 89.19 |
| MoG | 0.71 | **90.66** | **96.44** | **71.60** | 91.55 | 94.27 | **88.09** | **91.18** | **91.85** | **89.46** |
| DoRA | 0.75 | 90.44 | **96.33** | 70.18 | 91.28 | 94.20 | 88.09 | 89.95 | 91.36 | 88.98 |
| DoRA(+MoG) | 0.75 | **90.63** | 96.10 | **71.51** | **91.34** | **94.33** | **88.81** | **90.93** | **91.99** | **89.46** |
| PiSSA | 0.71 | 90.21 | **96.22** | 69.97 | 90.88 | 93.48 | **87.72** | 90.93 | 90.93 | 88.79 |
| PiSSA(+MoG) | 0.71 | **90.37** | 96.10 | **70.96** | **91.16** | **93.91** | 85.92 | **91.42** | **91.68** | **88.94** |

play a crucial role in effectively combining different bases. Furthermore, when the gating network is removed, the average further drops to 84.9% (–1.4%), indicating that the gating mechanism can adaptively select different bases based on input features, thereby enhancing the model's expressive capability and task adaptability. These results clearly demonstrate that both components are indispensable: gating provides dynamic flexibility, while cross-mixing expands the representational subspace. Their synergy accounts for MoG's superior performance on challenging reasoning tasks.

## 5.4 Scalability Analysis

**Parameter Efficiency Analysis.** We assess MoG's robustness under different parameter budgets by varying the rank $r$, with results provided in Table 11. As $r$ decreases from 220 to 16, the proportion of trainable parameters drops from 0.70% to 0.04%, yet performance declines by only 2.0 points (86.3% → 84.3%). Notably, even at $0.04\%$, MoG still surpasses LoRA (84.3% vs. 80.8%), highlighting superior parameter efficiency. Furthermore, comparisons with other methods under varying budgets (Figure 3) show that MoG consistently achieves higher accuracy at comparable or lower parameter counts, confirming its robustness across parameter budgets.

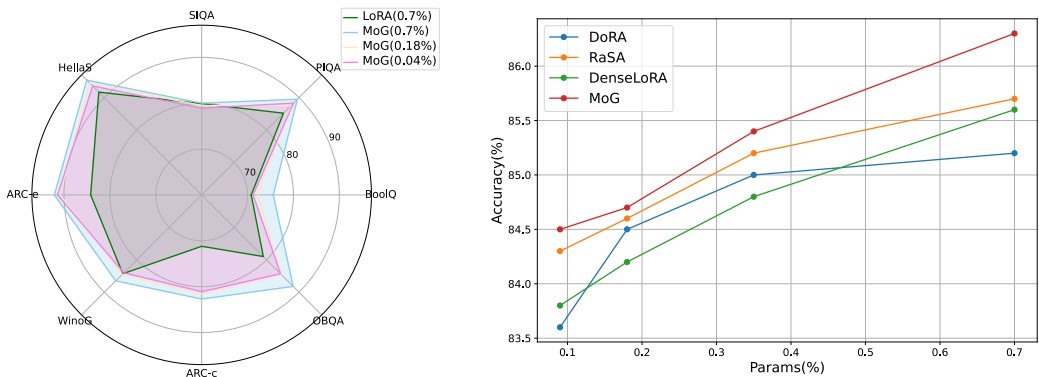

Figure 3: Performance on commonsense reasoning under varying parameter budgets.

**Impact of the Number of Groups on Performance.** We next study how the group number $n$ affects model performance, with results shown in Tables 12, 13 and Figures 4, 8. A consistent trend is observed for both DeBERTa-V3 (12 layers) and LLaMA3-8B (32 layers): performance initially improves as $n$ increases, peaks at a medium value, and then gradually declines. This phenomenon indicates that the optimal number of groups correlates with model depth but generally peaks at medium-sized groupings. When $n$ is too small, excessive sharing within groups leads to convergence of updates between layers, diminishing each layer's ability to process specific tasks. Conversely, when $n$ is too large, inter-layer information exchange is weakened, reducing both representation quality and training efficiency. In summary, medium-sized groupings ($n = 3 \sim 4$) achieve the best trade-off between parameter sharing and layer differentiation, validating MoG's design principle of balancing efficiency with expressiveness.

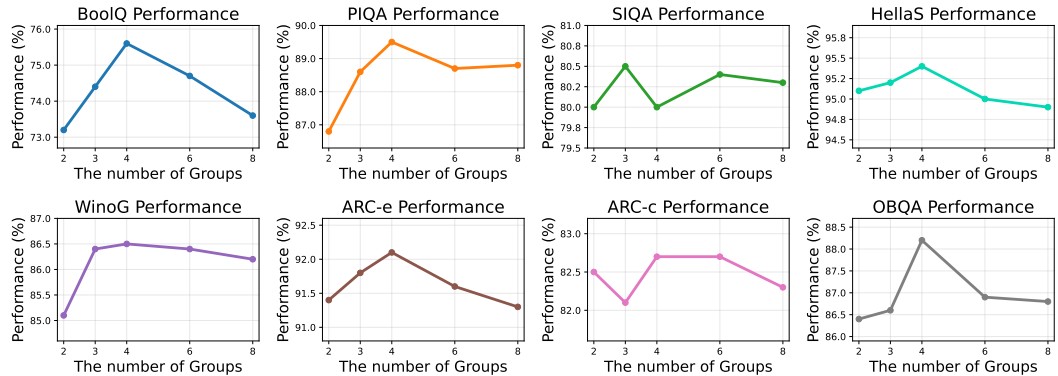

Figure 4: Impact of the number of groups on commonsense reasoning performance in LLaMA3-8B.

**Gating Network Visualization.** To better understand how MoG allocates parameters across groups, we visualize the learned gating coefficients for different layers, shown in Figure 5. The gating network is initialized with one-hot patterns, resulting in the assigned group dominating at the beginning, but after training, the distribution becomes softer, showing that the network allows cross-group interactions rather than remaining fixed. Moreover, small numerical differences appear between tasks, indicating that while the gating network preserves a stable global structure, it also fine-tunes the relative strengths of different groups according to task requirements. These results confirm that the gating mechanism is both adaptive and robust, supporting effective parameter sharing without losing task-specific flexibility.

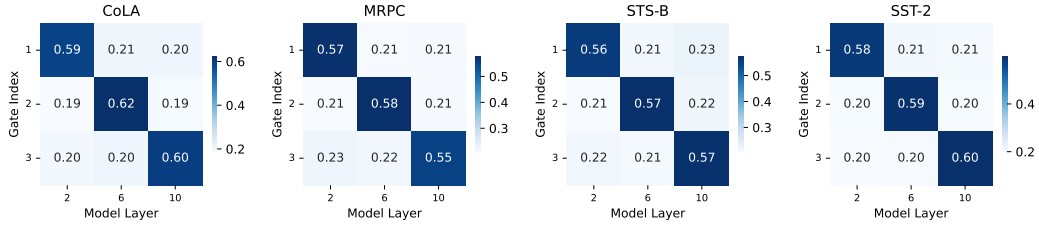

Figure 5: Visualization of normalized gating coefficients on DeBERTa-V3 for GLUE tasks.

**Memory Consumption and Training Time.** Table 14 compares GPU memory usage and relative training time on LLaMA3-8B. MoG consumes 41.38 GB, slightly lower than LoRA (42.01 GB). In terms of efficiency, MoG introduces a modest 8% training-time overhead relative to LoRA, aligning closely with RaSA. The additional cost arises mainly from the gating and cross-mixing computations. This demonstrates that while MoG introduces a slight overhead, it significantly enhances performance across multiple tasks and remains feasible for practical training and deployment.

## 6 CONCLUSION

In this work, we propose MoG, a novel PEFT framework that leverages the inherent similarity between adjacent layers to balance parameter sharing and layer-specific flexibility. By integrating group-wise low-rank sharing, adaptive gated aggregation, and cross-mixing, MoG effectively reduces redundancy while maintaining essential inter-layer diversity. Extensive experiments on four benchmarks demonstrate clear improvements over prior approaches. Overall, MoG provides a unified and flexible paradigm for parameter-efficient fine-tuning. Future work will focus on applying MoG to larger-scale foundation models and exploring its integration with other adaptation techniques to further broaden its applicability.

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

## A    THE USE OF LARGE LANGUAGE MODELS

We used ChatGPT for grammar checking and language polishing to improve the academic style and clarity of the manuscript. All content generated by the LLM was carefully reviewed and corrected by the authors to ensure accuracy and faithfulness to the original meaning. The authors take full responsibility for the final content of this paper.

## B    TRAINING DETAILS

For all experiments, we keep the total number of trainable parameters approximately the same across methods to ensure a fair comparison. Ranks are adjusted accordingly for different methods. All experiments are conducted on a single NVIDIA A6000 GPU (48GB memory).

### B.1    COMMONSENSE REASONING

We fine-tune LLaMA3-8B model with LoRA rank 32 (0.7% of trainable parameters). For MoG, we adopt a rank of 220 with the number of groups $n = 4$. For RaSA, the rank is set to 32 with $rasa\_k = 4$, and DenseLoRA is configured with a rank of 456. This setup ensures fair comparison among different methods under approximately the same parameter count.

Table 7: Hyperparameters for commonsense reasoning.

| Hyperparameter | Value |
|---|---|
| Base Model | Llama3-8B |
| Target Modules | $[W_q, W_k, W_v, W_{up}, W_{down}]$ |
| Epochs | 3 |
| Learning Rate | 1e-4 |
| Dropout | 0.05 |
| Optimizer | AdamW |
| Batch Size | 4 |
| Gradient Accumulation Steps | 4 |
| LR Scheduler | Linear |

### B.2    MATHEMATICAL REASONING AND CODE GENERATION

For mathematical reasoning and code generation tasks, we conduct fine-tuning experiments on Llama2-7B and Llama3-8B backbones. The rank of LoRA is set to 32, while for other approaches, the ranks are adjusted following the same strategy as described in the commonsense reasoning experiments.

Table 8: Hyperparameters for mathematical reasoning and code generation.

| Hyperparameter | Value |
|---|---|
| Base Model | [Llama2-7B, Llama3-8B] |
| Target Modules | $[W_q, W_k, W_v, W_o, W_{gate}, W_{up}, W_{down}]$ |
| Epochs | 2 |
| Learning Rate | [5e-5, 1e-4] |
| Dropout | 0.05 |
| Optimizer | AdamW |
| Batch Size | 4 |
| Gradient Accumulation Steps | 4 |
| LR Scheduler | Cosine |

### B.3 GLUE BENCHMARK

We fine-tune the DeBERTa-V3 model on the GLUE benchmark. The rank of LoRA is set to 8. For MoG, the number of groups is set to 3 with a rank of 30. DenseLoRA is configured with a rank of 70, and VBLoRA uses a rank of 8 with the number of vectors 244. A uniform batch size of 32 is used for all experiments. The fine-tuning target modules are ['query_proj', 'key_proj', 'value_proj', 'output.dense', 'intermediate.dense'].

Table 9: Hyperparameters for the GLUE Benchmark.

| Dataset | Epochs | Batch Size | Learning Rate |
|---------|--------|------------|---------------|
| MNLI    | 5      | 32         | 2e-4          |
| SST-2   | 5      | 32         | 2e-4          |
| MRPC    | 10     | 32         | 8e-4          |
| CoLA    | 10     | 32         | 2e-4          |
| QNLI    | 5      | 32         | 2e-4          |
| QQP     | 5      | 32         | 2e-4          |
| RTE     | 12     | 32         | 5e-4          |
| STS-B   | 10     | 32         | 1e-3          |

## C PROOF OF MOG'S EXPRESSIVE POWER

In this section, we give the details of the proof of the expressive power of MoG in comparison to LoRA. Our structure theoretically constitutes a larger matrix subspace, endowed with enhanced capability to approximate arbitrary matrices. Expanding Eq. 7, the layer update is

$$
\begin{aligned}
\Delta W_i &= \bar{A}_i \cdot E_i \cdot \bar{B}_i \\
&= \left( \sum_{g=1}^{n} \alpha_{ig} \cdot A_g \right) \cdot E_i \cdot \left( \sum_{g=1}^{n} \beta_{ig} \cdot B_g \right) \\
&= \sum_{g=1}^{n} \sum_{g'=1}^{n} \alpha_{ig} \beta_{ig'} \cdot A_g \cdot E_i \cdot B_{g'}.
\end{aligned}
\tag{8}
$$

Each term $A_g E_i B_{g'}$ has rank at most $r$, so $\Delta W_i$ is a weighted combination of $n^2$ rank-$r$ matrices. Let the subspaces spanned by $\{A_g\}$ and $\{B_{g'}\}$ be

$$
\begin{aligned}
\mathcal{A} &= \text{span}\{A_1, \ldots, A_n\}, \\
\mathcal{B} &= \text{span}\{B_1, \ldots, B_n\}.
\end{aligned}
\tag{9}
$$

The constructible tensor representation space is:

$$
\mathcal{F}_i = \{A \cdot E_i \cdot B \mid A \in \mathcal{A}, \ B \in \mathcal{B}\}.
\tag{10}
$$

This indicates that the matrix space of our method is a linear combination of tensor families $\mathcal{F}_i$, that is a matrix subspace of rank no greater than $n^2 \cdot r$.

In comparison, the expressive space of LoRA is

$$
\mathcal{S}_{\text{LoRA}} = \{AB \mid A \in \mathbb{R}^{d \times r}, B \in \mathbb{R}^{r \times k}\}.
\tag{11}
$$

The space constructed by MoG is

$$
\mathcal{S}_{\text{MoG}} = \left\{ \sum_{g,g'} \alpha_{ig} \beta_{ig'} A_g E_i B_{g'} \right\}.
\tag{12}
$$

Since LoRA corresponds to the special case $n = 1$ and $E_i = I_r$, it follows that

$$
\mathcal{S}_{\text{LoRA}} \subseteq \mathcal{S}_{\text{MoG}}.
\tag{13}
$$

Hence, MoG strictly generalizes LoRA: it reduces to LoRA in the extreme case, but in general spans a larger subspace with up to $n^2 r$ degrees of freedom.

# D ADDITIONAL LAYER SIMILARITY ANALYSIS

In the paper, we analyze inter-layer similarity on LLaMA2-7B using Distributed Cosine Similarity (DOCS)(Min & Wang, 2025), showing that adjacent layers of LoRA often exhibit high similarity and clustering tendencies. To further validate this phenomenon across different architectures, we extend the analysis to LLaMA3-8B and DeBERTa-V3.

Figures 6, 7 present the similarity heatmaps and t-SNE visualizations for the two models. Consistent with the observations on LLaMA2-7B, both LLaMA3-8B and DeBERTa-V3 display inter-layer correlations, where adjacent layers tend to cluster together. This supports the hypothesis that redundancy and structural similarity are general properties of Transformer-based models, and not specific to a single architecture. These findings further motivate our group-based parameter sharing strategy in MoG.

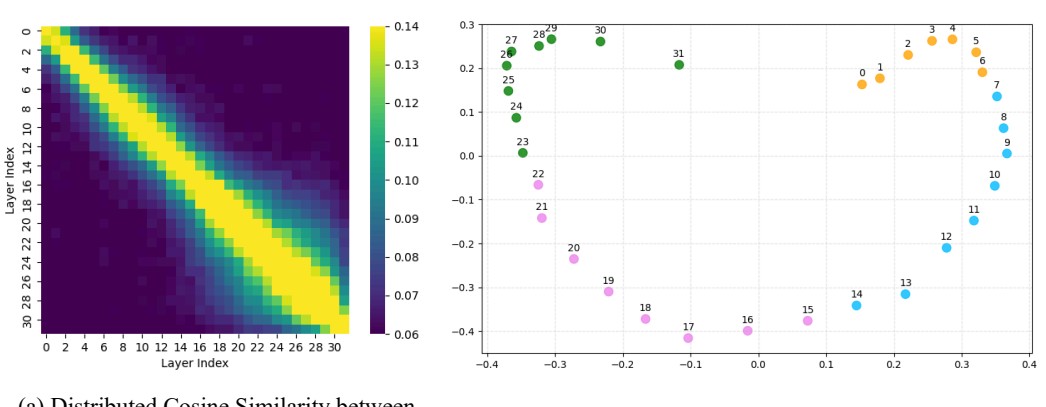

(a) Distributed Cosine Similarity between different LoRA layers

(b) t-SNE visualization of LoRA weights

Figure 6: Layer similarity visualizations on LLaMA3-8B.

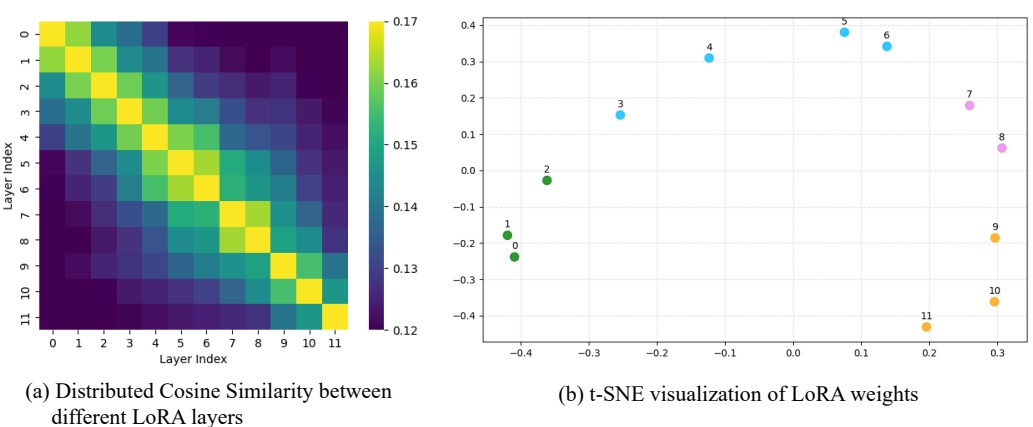

(a) Distributed Cosine Similarity between different LoRA layers

(b) t-SNE visualization of LoRA weights

Figure 7: Layer similarity visualizations on DeBERTa-V3.

# E ABLATION RESULT

We conduct ablation experiments on the commonsense reasoning benchmark to explore the role of different components, with the results shown in Table 10.

Table 10: Performance impact of removing architectural components from MoG. "w/o mixing weight" denotes the variant without the mixing weight matrix. "w/o gate" denotes the variant without the gating network.

| Method | # Params (%) | BoolQ | PIQA | SIQA | HellaS | WinoG | ARC-e | ARC-c | OBQA | Avg. |
|---|---|---|---|---|---|---|---|---|---|---|
| MoG | 0.70 | 75.6 | 89.5 | 80.0 | 95.4 | 86.5 | 92.1 | 82.7 | 88.2 | **86.3** |
| w/o mixing weight | 0.70 | 73.4 | 88.8 | 80.3 | 94.7 | 85.8 | 91.2 | 80.2 | 86.4 | 85.1 |
| w/o gate | 0.70 | 72.6 | 88.6 | 80.0 | 94.5 | 86.0 | 91.0 | 80.7 | 85.8 | 84.9 |

# F  RESULTS OF FURTHER EXPERIMENTS

## F.1  PARAMETER EFFICIENCY RESULT

We evaluate the parameter efficiency of MoG under different ranks, with the results shown in Table 11.

Table 11: Parameter efficiency of MoG at different ranks on the commonsense reasoning benchmark.

| Method | # Params (%) | BoolQ | PIQA | SIQA | HellaS | WinoG | ARC-e | ARC-c | OBQA | Avg. |
|---|---|---|---|---|---|---|---|---|---|---|
| MoG(r=220) | 0.70 | 75.6 | 89.5 | 80.0 | 95.4 | 86.5 | 92.1 | 82.7 | 88.2 | 86.3 |
| MoG(r=118) | 0.35 | 73.8 | 88.6 | 80.1 | 95.0 | 85.4 | 91.6 | 81.8 | 86.8 | 85.4 |
| MoG(r=64) | 0.18 | 72.4 | 87.6 | 79.8 | 94.5 | 84.5 | 91.0 | 81.6 | 86.5 | 84.7 |
| MoG(r=32) | 0.09 | 72.1 | 87.3 | 79.3 | 94.3 | 84.0 | 91.5 | 80.9 | 86.2 | 84.5 |
| MoG(r=16) | 0.04 | 71.2 | 88.3 | 78.9 | 93.6 | 83.6 | 91.4 | 81.1 | 86.2 | 84.3 |

## F.2  IMPACT OF THE NUMBER OF GROUPS ON PERFORMANCE

We investigate the impact of varying group sizes on performance, with results presented in Tables 12 and 13. Visualisation results are shown in Figures 4 and 8.

Table 12: Performance on the commonsense reasoning benchmark with varying group numbers (n).

| n | BoolQ | PIQA | SIQA | HellaS | WinoG | ARC-e | ARC-c | OBQA | Avg. |
|---|---|---|---|---|---|---|---|---|---|
| 2 | 73.2 | 86.8 | 80.0 | 95.1 | 85.1 | 91.4 | 82.5 | 86.4 | 85.1 |
| 3 | 74.4 | 88.6 | 80.5 | 95.2 | 86.4 | 91.8 | 82.1 | 86.6 | 85.7 |
| **4** | 75.6 | 89.5 | 80.0 | 95.4 | 86.5 | 92.1 | 82.7 | 88.2 | **86.3** |
| 6 | 74.7 | 88.7 | 80.4 | 95.0 | 86.4 | 91.6 | 82.7 | 86.9 | 85.8 |
| 8 | 73.6 | 88.8 | 80.3 | 94.9 | 86.2 | 91.3 | 82.3 | 86.8 | 85.5 |

Table 13: Performance on the GLUE benchmark with varying group numbers (n).

| n | MNLI | SST-2 | CoLA | QQP | QNLI | RTE | MRPC | STS-B | Avg. |
|---|---|---|---|---|---|---|---|---|---|
| 2 | 90.42 | 96.33 | 70.59 | 91.35 | 93.99 | 87.28 | 90.93 | 91.96 | 89.11 |
| **3** | 90.66 | 96.44 | 71.60 | 91.55 | 94.27 | 88.09 | 91.18 | 91.85 | **89.46** |
| 4 | 90.28 | 95.99 | 71.20 | 91.25 | 94.07 | 87.64 | 91.18 | 91.69 | 89.16 |
| 6 | 90.25 | 96.10 | 70.81 | 91.19 | 93.87 | 87.14 | 91.06 | 91.77 | 89.02 |

## F.3  MEMORY CONSUMPTION AND TRAINING TIME

We conduct experiments on a single NVIDIA A6000 GPU (48GB memory). Memory consumption and training time are shown in Table 14.

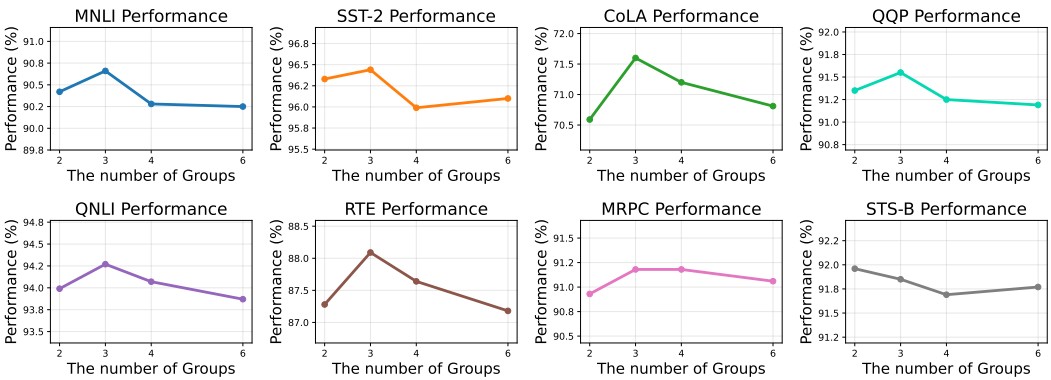

Figure 8: Impact of the number of groups on the GLUE benchmark performance in DeBERTa-V3.

Table 14: GPU memory consumption and training time on LLaMA3-8B with $r = 32$.

| Method | GRAM(GB) | Training Time($\times$) |
| --- | --- | --- |
| LoRA | 42.01 | 1.00$\times$ |
| DenseLoRA | 41.14 | 0.99$\times$ |
| RaSA | 42.36 | 1.09$\times$ |
| MoG | 41.38 | 1.08$\times$ |

## G  DETAILED PARAMETER COMPLEXITY ANALYSIS

We provide concrete parameter counts using a 32-layer model with hidden dimension $d = k = 1024$, adapter rank $r = 8$, and number of groups $n = 4$.

**LoRA.**

$$P_{\text{LoRA}} = L \cdot r(d + k) = 32 \cdot 8 \cdot (1024 + 1024) = 32 \cdot 8 \cdot 2048 = 524{,}288.$$

**MoG.**  The total parameters are

$$P_{\text{MoG}} = nr(d + k) + Lr^2 + Ln.$$

Substituting values:

$$\begin{aligned}
P_{\text{MoG}} &= 4 \cdot 8 \cdot (1024 + 1024) + 32 \cdot 8^2 + 32 \cdot 4 \\
&= 65{,}536 + 2{,}048 + 128 \\
&= 67{,}712.
\end{aligned}$$

**Comparison.**  MoG requires about 67.7k parameters compared to LoRA's 524.3k, an $\approx 87\%$ reduction in parameter count.

## H  CLUSTER ANALYSIS

To understand the intrinsic behavior of the model, we compute layer-wise similarity and perform clustering on both the pre-trained and fine-tuned model weights (Figure 9). Our analysis reveals two key observations:

- Layers naturally form several coherent clusters in the pre-trained model, indicating that adjacent layers often share similar functional behavior (Figure 9(a)).

- This clustered structure remains largely unchanged after fine-tuning, demonstrating that the grouping pattern is an intrinsic and stable property of the model (Figure 9(b)).

These findings motivate the structure-aware grouping strategy in MoG. Moreover, the cluster analysis offers a practical guideline for selecting the optimal number of groups. The pre-trained model's layers approximately form four clusters. Our experiments also show that using four groups achieves the best performance on commonsense benchmark for LLaMA3-8B (Sec.5.4), aligning closely with the empirical cluster structure. This alignment provides strong evidence that structure-aware grouping is both meaningful and effective.

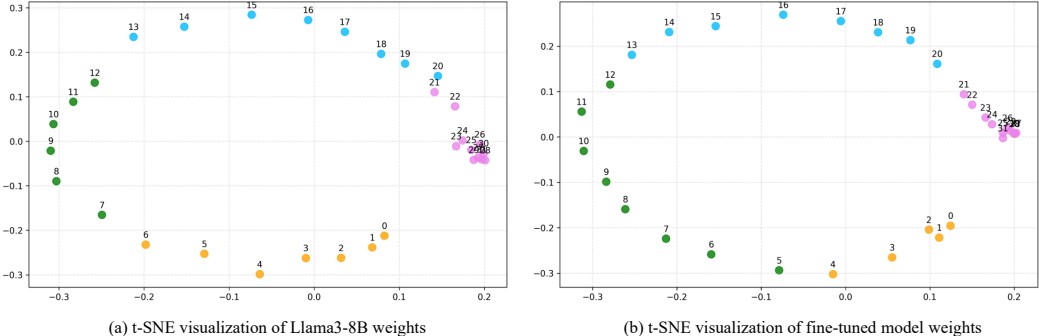

(a) t-SNE visualization of Llama3-8B weights         (b) t-SNE visualization of fine-tuned model weights

Figure 9: t-SNE visualizations of the pre-trained and fine-tuned model weights from LLaMA3-8B.

