# OpenReview forum: "Mixture of Groups: Grouped Gating and Cross Mixing for Parameter-Efficient LLM Fine-Tuning"
_ICLR.cc/2026/Conference — Submitted to ICLR 2026_

### Official Review · Reviewer_buem · 2025-10-23

**Soundness:** 3
**Presentation:** 3
**Contribution:** 3
**Rating:** 6
**Confidence:** 4

**Summary:**

This paper proposes Mixture of Groups (MoG), a new parameter-efficient fine-tuning (PEFT) framework for large language models. It clusters adjacent layers into functionally similar groups that share low-rank adapters and augments them with adaptive gating plus cross-mixing, breaking the dichotomy between per-layer LoRA and fully shared approaches. Extensive experiments on several LLMs and benchmarks show that MoG delivers higher accuracy than previous PEFT baselines under the same or smaller parameter budgets.

**Strengths:**

1. Thoughtful design: grouping, gating, and cross-mixing are well-motivated by empirical analyses of inter-layer similarity and provide a principled middle ground between fully shared and fully independent adapters.

2. The orthogonal design enables MoG to benefit from a wide range of LoRA variants.

3. The paper is well written and easy to follow.

**Weaknesses:**

1. Comparative puzzle: vanilla LoRA underperforms DoRA in your tables, yet LoRA+MoG surpasses DoRA+MoG; please clarify why MoG benefits LoRA more than DoRA—does MoG overlap with DoRA’s decomposition or were hyper-parameters unequal?

2. From my experience, llm-adapter framework is very sensitive to learning rates in the range {1e-4, 2e-4, 3e-4}. The vanilla LoRA result on Llama-3 8B was likely obtained with a learning rate of 3e-4, whereas the other methods were not. Could you rerun LoRA on Llama-3 8B with a 1e-4 learning rate and report the results to provide readers with a clearer comparison?

**Questions:**

please see weakness

---

> ### Author Response · Authors · 2025-11-20
> **Response to Reviewer buem**
>
> We sincerely thank you for the detailed and insightful comments. Our responses are provided below.
>
> > **Weakness 1.** Why does MoG benefit LoRA more than DoRA?
>
> **Response:** This arises from the structural characteristics of LoRA and DoRA, rather than from hyperparameter differences. DoRA decomposes weights into direction and magnitude components, which already reduces redundancy and increases representational expressivity. LoRA, in contrast, applies low-rank updates directly and therefore retains more redundancy in the adapter weights. Since MoG is specifically designed to leverage redundancy and similarity across layers, it naturally yields larger gains for LoRA.
>
>
>
> > **Weekness 2.** Clarification of the LoRA baseline
>
> **Response:** You pointed out that the LLM-adapter framework is sensitive to learning rates in the range {1e-4, 2e-4, 3e-4}. The vanilla LoRA result on Llama-3 8B (80.8) reported in our paper is taken from the DoRA paper (Zhang et al., 2024)[1] , which uses a learning rate of 1e-4. Many recent works (e.g., HiRA [2], DenseLoRA [3]) also adopt the same baseline. To provide readers with a clearer comparison, we have rerun LoRA on Llama-3 8B with the following learning rates.
>
> |                | BoolQ | PIQA | SIQA | HellaS | WinoG | ARC-e | ARC-c | OBQA | Avg. |
> | -------------- | ----- | ---- | ---- | ------ | ----- | ----- | ----- | ---- | ---- |
> | LoRA (lr=1e-4) | 73.3  | 88.6 | 79.5 | 91.1   | 86.1  | 84.4  | 74.2  | 81.6 | 82.4 |
> | LoRA (lr=2e-4) | 71.4  | 86.2 | 80.3 | 92.0   | 85.7  | 87.3  | 75.8  | 84.0 | 82.8 |
> | LoRA (lr=3e-4) | 68.8  | 84.5 | 77.0 | 67.8   | 82.7  | 85.9  | 73.4  | 82.0 | 77.8 |
>
> The results reflect the performance under our training settings, and we provide them here for completeness and transparency.
>
>
>
> **References**
>
> [1] Liu S Y, Wang C Y, Yin H, et al. Dora: weight-decomposed low-rank adaptation. ICML2024.
>
> [2] Qiushi Huang, Tom Ko, Zhan Zhuang, et al. HiRA: Parameter-efficient hadamard high-rank adaptation for large language models. ICLR2025.
>
> [3] Lin Mu, Xiaoyu Wang, Li Ni, et al. Denselora: Dense low-rank adaptation of large language models. ACL2025.

---

### Official Review · Reviewer_Cvwa · 2025-10-25

**Soundness:** 3
**Presentation:** 3
**Contribution:** 2
**Rating:** 4
**Confidence:** 4

**Summary:**

This paper introduces MoG, a novel framework for PEFT of LLMs. The paper identifies a key limitation in the popular LoRA method: by learning independent adapters for every layer, LoRA ignores the functional similarity between adjacent layers, leading to parameter redundancy. Conversely, other methods that rely on global parameter sharing can overlook layer-specific functional diversity, thereby limiting expressiveness.

**Strengths:**

- Balanced Framework: MoG introduces a novel group-sharing framework that balances parameter efficiency by reducing redundancy with layer-specific expressiveness.
- Flexible Architecture: The model uses adaptive gating and cross-mixing to create a highly flexible and expressive adaptation subspace that generalizes LoRA.
- Versatile Plug-and-Play Module: MoG is designed as a general-purpose module that can be plugged into other PEFT methods, such as DoRA and PiSSA, to boost their performance.
- Strong Empirical Results: The method is validated by extensive experiments across multiple LLMs and benchmarks, demonstrating superior performance over strong baselines with comparable parameter budgets.

**Weaknesses:**

- `Limited Motivation and Static Mechanism`: The paper claims to employ adaptive gating and cross-mixing mechanisms to capture both commonalities and unique features across layers by leveraging inter-layer similarity. However, a significant limitation is that the fusion weights generated for each layer or group are static. This approach appears disconnected from the paper's own findings, as illustrated in Figure 1(a), which clearly indicates dynamic variations in cosine similarity between different layers. The potential relationship between these dynamic similarity scores and the static fusion weights is a critical, unexplored area that warrants further investigation.
- `Limited Comparative Analysis`: The paper's methodology draws inspiration from MoE principles to some extent. Despite this, the experimental evaluation conspicuously omits comparisons against relevant and contemporary MoE-based LoRA variants ($e.g.$, [1-3]) as baselines. I recommend incorporating these methods to provide a truly comprehensive and meaningful comparison. Furthermore, the experimental scope is narrow regarding model scale and architecture. The analysis would be substantially strengthened by including evaluations on models of varying sizes, such as Qwen2.5-3B, Qwen2.5-7B, and Qwen2.5-14B, to demonstrate the method's scalability.
- `Limited Application Scenarios`: The empirical validation is currently confined to single-task environments. This narrow focus significantly limits the method's potential applicability and generalizability, particularly for the increasingly prevalent and complex multi-task scenarios found in real-world applications. To substantiate the robustness and effectiveness of the proposed approach, it is essential to extend the evaluation to include challenging multi-task benchmarks ($e.g.$, OpenOrca).

[1] When MOE Meets LLMs: Parameter Efficient Fine-tuning for Multi-task Medical Applications

[2] HydraLoRA: An Asymmetric LoRA Architecture for Efficient Fine-Tuning

[3] CoLA: Collaborative Low-Rank Adaptation

**Questions:**

See Weaknesses.

---

> ### Author Response · Authors · 2025-11-20
> **Response to Reviewer Cvwa, Part1**
>
> We sincerely thank you for the constructive comments. In the following, we respond to each of the concerns raised.
>
> > **Weekness 1.** Limited Motivation and Static Mechanism
>
> **Response:** Our grouping strategy is not arbitrary. It is directly guided by **clustering the pre-trained model’s weights**, revealing which layers share close functional behavior and which layers are distinct. Although layer similarity may fluctuate during fine-tuning, the structural of the pre-trained model provides a stable and reliable basis for grouping. Previous work such as DOCS (Min et al., 2025)[1] shows the base and instruction-tuned models retain a high degree of similarity after the fine-tuning process, providing support for using pre-trained similarity patterns to guide grouping. Our experiments confirm that these cluster-based groups are effective. As shown in Figure 1(b), the layers approximately form 4 clusters, which is consistent with the analysis in Section 5.4 of our paper, where we find that using **3-4 groups** results in the **best performance**. This alignment between clustering patterns and empirical results demonstrates that our grouping strategy works well in practice.
>
>
>
> > **Weakness 2.** Limited Comparative Analysis
>
> **Response:** MoG offers several advantages compared with MoE-based LoRA methods:
>
> - **Inter-layer structure modeling:** MoG leverages functional similarity across layers to guide group sharing, reducing redundancy while preserving layer-specific specialization. In contrast, MoE-LoRA methods operate only within individual layers and do not model cross-layer structure.
> - **Higher parameter efficiency:** MoG replaces full LoRA modules with group-shared modules, making it much more parameter-efficient than MoE-LoRA, which requires multiple experts per layer, increasing parameter count and computation.
> - **Global information integration:** MoG employs a lightweight gating network to aggregate information across all groups, allowing each layer to adaptively incorporate global signals. In contrast, MoE-based LoRA methods treat each layer independently and do not provide a mechanism to share global information. Moreover, their routing mechanism becomes increasingly expensive as the number of experts grows, adding additional overhead.
>
> In addition, MoE-based LoRA methods (e.g., MoELoRA[2], CoLA[3]) are designed mainly to enhance multi-task learning capacity within each layer, rather than to improve inter-layer parameter efficiency. Since their objectives and mechanisms differ from MoG, the comparison is not entirely direct. Nevertheless, to provide a more complete picture, we have conducted additional experiments on the commonsense reasoning benchmark, comparing MoG with MoE-LoRA variants, including CoLA[3] and HydraLoRA[4].
>
> | **Method** | **# Params (%)** | **BoolQ** | **PIQA** | **SIQA** | **HellaS** | **WinoG** | **ARC-e** | **ARC-c** | **OBQA** | **Avg.** |
> | ---------- | ---------------- | --------- | -------- | -------- | ---------- | --------- | --------- | --------- | -------- | -------- |
> | HydraLoRA  | 0.70             | 72.7      | 89.4     | **80.3** | 95.1       | 86.4      | 90.3      | 79.8      | 86.0     | 85.0     |
> | CoLA       | 0.74             | 73.2      | 88.5     | 80.2     | 95.2       | 86.2      | 91.2      | 80.1      | 86.6     | 85.2     |
> | MoG        | 0.35             | 73.8      | 88.6     | 80.1     | 95.0       | 85.4      | 91.6      | 81.8      | 86.8     | 85.4     |
> | MoG        | 0.70             | **75.6**  | **89.5** | 80.0     | **95.4**   | **86.5**  | **92.1**  | **82.7**  | **88.2** | **86.3** |
>
> The results demonstrate that with only half the parameter budget (0.35%), MoG outperforms HydraLoRA and CoLA, confirming that MoG delivers parameter efficiency and effectiveness than MoE-based LoRA methods.
>
> Given limited compute resources, we have also conducted additional experiments on the large-scale **Qwen2.5‑14B** model to further evaluate the scalability of MoG. We fine-tuned on the Metamath and CodeFeedback datasets for one epoch. The results show that MoG achieves the best performance across all four test sets.
>
> | **Method** | **# Params (%)** | **GSM8K** | **MATH** | **HumanEval** | **MBPP** |
> | ---------- | ---------------- | --------- | -------- | ------------- | -------- |
> | LoRA       | 0.92             | 79.5      | 50.2     | 76.6          | 83.9     |
> | RaSA       | 0.92             | 81.4      | 51.0     | 78.4          | 84.1     |
> | DenseLoRA  | 0.92             | 80.8      | 50.6     | 77.8          | 84.7     |
> | MoG        | 0.92             | **82.3**  | **51.2** | **79.9**      | **85.2** |

---

> ### Author Response · Authors · 2025-11-20
> **Response to Reviewer Cvwa, Part2**
>
> > **Weakness 3.** Limited Application Scenarios
>
> **Response:** Our expeiments have **covered commonsense reasoning, math & code tasks, and NLU benchmark**. These tasks provide a comprehensive evaluation of the model's capabilities. In particular, our task selection follows widely adopted settings in prior works such as PiSSA [5], HiRA [6], RaSA [7], and DenseLoRA [8], enabling a fair and comparable evaluation framework.
>
> We agree that broader multi-task evaluation would further strengthen the study. We appreciate this suggestion and have conducted **additional multi-task experiments**. Specifically, We fine-tuned Llama3-8B on **OpenOrca-35k** [9], a widely used multi-task instruction-tuning dataset, and evaluated on **MMLU** [10], which includes 57 diverse sub-domains across humanities, social science, STEM, and other professional fields. The results are shown below:
>
>
>
> | **Method** | **# Params (%)** | **Hums.** | **Social** | STEM      | **Other** | Avg.      |
> | ---------- | ---------------- | --------- | ---------- | --------- | --------- | --------- |
> | LoRA       | 0.70             | 58.94     | 74.08      | 55.90     | 72.48     | 65.35     |
> | DoRA       | 0.71             | 59.06     | 74.56      | 56.67     | 72.68     | 65.74     |
> | RaSA       | 0.70             | 59.46     | 74.29      | **56.98** | 73.06     | 65.95     |
> | DenseLoRA  | 0.70             | 57.68     | 74.29      | 55.19     | 72.45     | 64.90     |
> | MoG        | 0.70             | **60.04** | **75.30**  | 56.90     | **73.35** | **66.40** |
>
> MoG achieves the highest overall MMLU score under the same parameter budget. These results validate that MoG generalizes well beyond single-task benchmarks and remains  highly competitive in multi-task fine-tuning scenarios. We believe this additional evidence further supports the robustness and applicability of MoG.
>
> **References**
>
> [1] Min Z, Wang X. Docs: quantifying weight similarity for deeper insights into large language models. ICLR2025.
>
> [2] Liu Q, Wu X, Zhao X, et al. When moe meets llms: Parameter efficient fine-tuning for multi-task medical applications. ACM SIGIR2024.
>
> [3] Zhou Y, Yao C, Chen J. Cola: Collaborative low-rank adaptation. arXiv:2505.15471.
>
> [4] Tian C, Shi Z, Guo Z, et al. Hydralora: An asymmetric lora architecture for efficient fine-tuning[J]. NIPS2024.
>
> [5] Meng F, Wang Z, Zhang M. Pissa: principal singular values and singular vectors adaptation of large language models. NIPS2024.
>
> [6] Qiushi Huang, Tom Ko, Zhan Zhuang, et al. HiRA: Parameter-efficient hadamard high-rank adaptation for large language models. ICLR2025.
>
> [7] He Z, Tu Z, Wang X, et al. Rasa: rank-sharing low-rank adaptation. ICLR2025.
>
> [8] Lin Mu, Xiaoyu Wang, Li Ni, et al. Denselora: Dense low-rank adaptation of large language models. ACL2025.
>
> [9] Lian W, Goodson B, Pentland E, et al. Openorca: An open dataset of gpt augmented flan reasoning traces. 2023.
>
> [10] Hendrycks D, Burns C, Basart S, et al. Measuring massive multitask language understanding. ICLR2021.

---

> > ### Comment · Reviewer_Cvwa · 2025-11-26
> >
> > Thank you for the response. I suggest restating the paper’s motivation and providing a more detailed explanation of the method (e.g., the clustering component) to make it clearer and more straightforward.

---

> > > ### Author Response · Authors · 2025-11-28
> > > **Response to Reviewer Cvwa**
> > >
> > > Thank you for the follow-up suggestion. Below we provide a clearer and more detailed explanation of both the motivation and the method.
> > >
> > > Standard LoRA learns a fully independent adapter at every layer, leading to redundancy. By exploiting the strong similarity among adjacent layers and grouping neighboring layers to share a single adapter, MoG reduces parameter redundancy while retaining the expressiveness. This design enables MoG to achieve significantly higher parameter efficiency than LoRA without sacrificing performance.
> > >
> > > Hard-sharing methods (e.g., RaSA[1], DenseLoRA[2], BSLoRA[3]) force all layers to share the same LoRA module. Such over-sharing ignores functional differences across layers and results in harming expressiveness. In contrast, MoG overcomes this limitation by introducing a learnable gating network and lightweight fusion matrix to adaptively aggregate information from different groups. These components enable each layer to access global knowledge while maintaining its unique behavior, effectively addressing the over-sharing problem.
> > >
> > > Together, MoG achieves a balanced trade-off between efficiency and expressiveness, outperforming both independent and hard-sharing PEFT strategies.
> > >
> > > To better understand the role of clustering, we added a cluster analysis in Appendix H. Comparing Fig. 9(a) and Fig. 9(b), we observe that adjacent layers naturally form several clusters and this structure remains stable, even after fine-tuning. The grouping pattern is nearly unchanged between pre-trained and instruction-tuned weights. This indicates that the model’s depth is organized into functionally coherent regions. Therefore, the cluster structure of the pre-trained model can provide a guideline for choosing the number of groups. As shown in Fig. 9(a), the layers approximately form four clusters, and our experiments also confirm that using four groups achieves the best performance on commonsense benchmark for LLaMA3-8B (Table 12). This alignment provides strong evidence that structure-aware grouping is both meaningful and effective.
> > >
> > >
> > >
> > > **References**
> > >
> > > [1] He Z, Tu Z, Wang X, et al. Rasa: rank-sharing low-rank adaptation. ICLR2025.
> > >
> > > [2] Lin Mu, Xiaoyu Wang, Li Ni, et al. Denselora: Dense low-rank adaptation of large language models. ACL2025.
> > >
> > > [3] Yuhua Zhou, Ruifeng Li, Changhai Zhou, et al. BSLoRA: Enhancing the parameter efficiency of loRA with intra-layer and inter-layer sharing. ICML2025.

---

### Official Review · Reviewer_p58V · 2025-10-31

**Soundness:** 3
**Presentation:** 3
**Contribution:** 3
**Rating:** 6
**Confidence:** 2

**Summary:**

Large language models (LLMs) excel in NLP but are costly for full fine-tuning. Low-Rank Adaptation (LoRA), a common Parameter-Efficient Fine-Tuning (PEFT) method, uses independent per-layer adapters, causing redundancy and wasting inter-layer similarity. We propose **Mixture of Groups (MoG)**, a new PEFT framework: it splits LLM layers into groups (sharing low-rank adapters to cut redundancy), uses adaptive gating for layer-specific parameter combination, and adds a cross-mixing module to boost representational capacity.

As a plug-and-play tool, MoG integrates with DoRA/PiSSA to enhance their performance. Experiments on commonsense reasoning, math, code generation, and GLUE show MoG outperforms LoRA/DenseLoRA under similar parameter budgets, balancing efficiency and effectiveness in LLM fine-tuning.

**Strengths:**

1.	Focusing on an important problem of existing PEFT methods: Clearly points out the limitations of LoRA (redundancy from independent per-layer adapters) and global sharing methods (e.g., VeRA, DenseLoRA, which ignore inter-layer functional diversity). It also verifies the functional similarity of adjacent LLM layers through experimental evidence like DOCS analysis and t-SNE visualization, providing solid theoretical and data support for the method design.
2.	Flexible architecture with strong compatibility: The MoG framework has a "plug-and-play" feature. It can be used independently as a PEFT method and seamlessly integrated into mainstream PEFT methods such as DoRA and PiSSA (e.g., DoRA(+MoG) and PiSSA(+MoG) both improve the performance of the original methods). It also supports architecture degradation (degenerates to LoRA when n=L and E is an identity matrix, and is similar to DenseLoRA when n=1), adapting to different fine-tuning needs.
3.	Modular component design with high interpretability: The three core components of MoG ("group sharing-adaptive gating-cross mixing") have clear functions and divisions of labor. Ablation experiments prove that no component is dispensable (removing the cross-mixing module reduces performance by 1.2%, and removing gating reduces it by 1.4%). Meanwhile, gating coefficient visualization shows that the gating distribution changes from "one-hot" to "flexible" after training, which dynamically adjusts the weight of group parameters according to tasks. The working mechanism of components is interpretable, facilitating subsequent optimization and improvement.
4.	Rigorous and comprehensive experimental design, with credible conclusions: Experiments cover multiple task types (reasoning, generation, understanding) and models of different scales. Baseline methods include mainstream PEFT methods like LoRA, RaSA, and DenseLoRA. A unified parameter budget ensures fair comparison. In addition, extended experiments (parameter efficiency analysis, group number impact analysis, resource consumption analysis) are added to verify the method’s advantages from different dimensions, making the conclusions more convincing.

**Weaknesses:**

1.	Limited exploration of group number selection. The optimal number of groups (n) varies by task/model, but no clear guidelines for choosing n are provided.
2.	Less testing on extremely large models. Most experiments use 7B/8B models; performance on larger models (e.g., 70B) remains unproven.
3.	Higher complexity than basic LoRA. The cross-mixing module and gating mechanism add minor computational overhead, though small.
4.	Limited analysis of hyperparameter sensitivity. How factors like rank (r) or learning rate affect MoG’s stability needs more study.

**Questions:**

1. Could you provide practical guidelines or automated methods to determine the optimal number of groups (n) for different tasks and model scales?
2. Have you tested MoG on larger models (e.g., 70B LLMs)? If not, do you expect similar performance gains there?
3. How sensitive is MoG to hyperparameters like rank (r) or learning rate? Are there stable ranges to recommend?

---

> ### Author Response · Authors · 2025-11-20
> **Response to Reviewer p58V**
>
> We sincerely thank you for your constructive and detailed comments. Below we provide a response addressing both the identified weaknesses and the questions.
>
> > **W1 & Q1.** Limited exploration of group number selection
>
> **Response:** A practical way to choose the number of groups *n* is to **cluster the pre-trained model’s weights** and use the resulting cluster count as the group number. Because this method can naturally reveal which layers share similar functional roles. Layers that consistently cluster together indicate high functional similarity and thus are suitable to be placed in the same group. In practice, we observe that the weights cluster into approximately **four groups**, as shown in Figure 1(b). Our experimental results support this: the analysis in our paper demonstrates that using 3–4 groups achieves the best performance.
>
> In summary, the optimal group number can be effectively determined from the distribution of the pre-trained model's weights, typically suggesting 3–4 groups for models in the 7B–8B range.
>
>
>
> > **W2 & Q2.** Limited testing on very large models
>
> **Response:** Due to computational constraints, our initial experiments focused on 7B/8B-scale models. To further evaluate the scalability of MoG, we have now conducted additional experiments on the **Qwen2.5-14B** model. These results show that MoG achieves the best performance across all four test sets, confirming that MoG remains highly effective as model size increases. We plan to expand our evaluation to 70B-scale models in future revisions when computational resources permit.
>
> | **Method** | **# Params (%)** | **GSM8K** | **MATH** | **HumanEval** | **MBPP** |
> | ---------- | ---------------- | --------- | -------- | ------------- | -------- |
> | LoRA       | 0.92             | 79.5      | 50.2     | 76.6          | 83.9     |
> | RaSA       | 0.92             | 81.4      | 51.0     | 78.4          | 84.1     |
> | DenseLoRA  | 0.92             | 80.8      | 50.6     | 77.8          | 84.7     |
> | MoG        | 0.92             | **82.3**  | **51.2** | **79.9**      | **85.2** |
>
>
>
> > **W3.** Higher complexity than basic LoRA
>
> **Response:** The computational overhead introduced by MoG is **minimal**. We measured the FLOPs on Llama3-8B. The base model's FLOPs amount to 1921.19G. At rank 32, LoRA's FLOPs are at 14.5G, while MoG's are 14.54G. The additional computation is only **0.04G FLOPs**. At the same time, MoG achieves **better parameter efficiency** than LoRA. When using a single group, MoG contains only 1.93M parameters, reducing **96.6%** of LoRA parameters (56.62M).
>
>
>
> > **W4 & Q3.** Sensitivity to hyperparameters (rank r and learning rate)
>
> **Response:**
>
> **Sensitivity to rank**
>
> We conducted a detailed study of the effect of different ranks on the Commonsense-170k benchmark (Table 11 in our paper). As shown in the results:
>
> | Rank | \# Params (%) | BoolQ | PIQA | SIQA | HellaS | WinoG | ARC-e | ARC-c | OBQA | Avg. |
> | ---- | ------------- | ----- | ---- | ---- | ------ | ----- | ----- | ----- | ---- | ---- |
> | 220  | 0.70          | 75.6  | 89.5 | 80.0 | 95.4   | 86.5  | 92.1  | 82.7  | 88.2 | 86.3 |
> | 118  | 0.35          | 73.8  | 88.6 | 80.1 | 95.0   | 85.4  | 91.6  | 81.8  | 86.8 | 85.4 |
> | 64   | 0.18          | 72.4  | 87.6 | 79.8 | 94.5   | 84.5  | 91.0  | 81.6  | 86.5 | 84.7 |
> | 32   | 0.09          | 72.1  | 87.3 | 79.3 | 94.3   | 84.0  | 91.5  | 80.9  | 86.2 | 84.5 |
> | 16   | 0.04          | 71.2  | 88.3 | 78.9 | 93.6   | 83.6  | 91.4  | 81.1  | 86.2 | 84.3 |
>
> As rank decreases from 220 to 16, performance declines by only 2.0 points. Even with very small ranks (only 0.04% trainable parameters), MoG retains strong performance (Avg. 84.3). These results indicate that MoG is robust to the rank, and practitioners can choose *r* within a broad range depending on parameter budget.
>
> **Sensitivity to Learning Rate**
>
> We further studied the effect of different learning rates. Our main experiments used a learning rate of 1e-4, which yields the best performance (86.3 average). We additionally ran experiments with learning rates 5e-5(85.6), 2e-4(85.4), and 3e-4(84.2). MoG shows stable performance within the range 5e-5 to 2e-4, with 1e-4 performing the best.
>
> | lr   | BoolQ | PIQA | SIQA | HellaS | WinoG | ARC-e | ARC-c | OBQA | Avg.     |
> | ---- | ----- | ---- | ---- | ------ | ----- | ----- | ----- | ---- | -------- |
> | 5e-4 | 74.7  | 89.0 | 80.7 | 95.0   | 85.3  | 92.2  | 81.2  | 86.8 | 85.6     |
> | 1e-4 | 75.6  | 89.5 | 80.0 | 95.4   | 86.5  | 92.1  | 82.7  | 88.2 | **86.3** |
> | 2e-4 | 74.0  | 88.8 | 80.1 | 95.1   | 85.6  | 91.8  | 81.6  | 86.4 | 85.4     |
> | 3e-4 | 73.0  | 87.2 | 79.6 | 94.1   | 85.6  | 88.9  | 79.2  | 85.6 | 84.2     |

---

> > ### Comment · Reviewer_p58V · 2025-11-24
> >
> > Thank you for the author's response. My score is already positive, and I will keep it.

---

### Official Review · Reviewer_NQti · 2025-10-31

**Soundness:** 2
**Presentation:** 3
**Contribution:** 2
**Rating:** 4
**Confidence:** 4

**Summary:**

This paper addresses the parameter redundancy problem in LoRA, which ignores the functional similarity between adjacent layers. It proposes a new PEFT framework called MoG. MoG reduces redundancy through group sharing, while using gated adaptive aggregation and cross mixing mechanisms to keep each layer’s specificity. This design achieves a delicate balance between parameter sharing and model expressiveness. Through extensive experiments, the authors show that MoG significantly outperforms LoRA and other sharing methods under the same parameter budget, achieving much higher parameter efficiency. Moreover, MoG is a general plug-and-play module that can be combined with other PEFT methods such as DoRA and PiSSA to further improve performance. Overall, MoG introduces an innovative and efficient new PEFT paradigm.

**Strengths:**

- The paper is well-written.
- The method is simple, and the plug-and-play design brings some performance gains.

**Weaknesses:**

- The paper lacks comparisons with related work. Many studies have optimized LoRA from the efficiency perspective, some focus on reducing the rank, while others focus on parameter sharing, but the baselines here do not include these methods. (https://arxiv.org/pdf/2406.10785 ; https://openreview.net/pdf?id=IXYBuwCOMl; https://arxiv.org/html/2410.11772v1 ; https://arxiv.org/html/2402.08562v1 ; https://arxiv.org/abs/2410.19694)
- The idea of "less is more" in PEFT has been discussed in many works (including but not limited to those mentioned above). Therefore, the contribution of this paper may be limited.
- The paper also lacks quantitative analysis of how much MoG actually reduces parameters. Since GPU memory usage mainly comes from the base model, the computational cost reduction of MoG may be limited.

**Questions:**

Please see weaknesses.

---

> ### Author Response · Authors · 2025-11-20
> **Response to Reviewer NQti, Part 1**
>
> We sincerely thank you for the constructive feedback. In the following, we respond to each of the points raised.
>
> > **Weakness 1.** Lack of comparisons with related PEFT methods.
>
> **Response:**   Existing parameter-sharing and pruning methods can indeed reduce redundancy and improve the efficiency of LoRA fine-tuning, but they still exhibit notable limitations. Hard-sharing approaches force all layers to use the same LoRA module (e.g., ShareLoRA[1], BSLoRA[2]), ignoring the substantial functional differences across layers and thus limiting the model’s ability to express layer-wise diversity. Dynamic pruning or merging methods (e.g., IST[3], XGBLoRA[4]) rely on importance scores or gradients to select parameters but lack cross-layer structural modeling, making it difficult to capture functional correlations between layers. In addition, MoE-based LoRA methods (e.g., MoLA[5], HydraLoRA[6]) focus on enhancing multi-task capability within each layer through expert routing, rather than improving efficiency through parameter sharing. Moreover, increasing the number of experts leads to higher routing overhead, which reduces their suitability in parameter-efficient settings.
>
> In contrast, MoG is **the first** PEFT method that **groups layers based on the model’s inherent functional structure**. By clustering the pre-trained weights, we can identify which layers exhibit high similarity for sharing, and which layers exhibit distinct behaviors to keep independent. Furthermore, MoG’s lightweight gating enables **the integration of global knowledge**. This design captures cross-group distinctions while adding almost no computational overhead. In addition, each layer only requires an independent  $r×r$ fusion matrix, which is **more efficient** than methods like BSLoRA, where every layer keeps a full LoRA module.
>
> To more clearly demonstrate MoG's advantages, **we have run additional experiments** comparing MoG with these methods. We selected a set of commonly used baselines: SharedLoRA[1], HydraLoRA[6], and BSLoRA[2] for comparison. The results are summarized below:
>
> | **Method** | **# Params (%)** | **BoolQ** | **PIQA** | **SIQA** | **HellaS** | **WinoG** | **ARC-e** | **ARC-c** | **OBQA** | **Avg.** |
> | ---------- | ---------------- | --------- | -------- | -------- | ---------- | --------- | --------- | --------- | -------- | -------- |
> | SharedLoRA | 0.73             | 72.1      | 88.0     | 79.8     | 94.5       | 85.6      | 90.1      | 79.6      | 85.6     | 84.4     |
> | HydraLoRA  | 0.70             | 72.7      | 89.4     | 80.3     | 95.1       | 86.4      | 90.3      | 79.8      | 86.0     | 85.0     |
> | BSLoRA     | 0.70             | 73.1      | 88.4     | **80.4** | 95.1       | 85.1      | 91.5      | 81.9      | 88.0     | 85.4     |
> | MoG        | 0.35             | 73.8      | 88.6     | 80.1     | 95.0       | 85.4      | 91.6      | 81.8      | 86.8     | 85.4     |
> | MoG        | 0.70             | **75.6**  | **89.5** | 80.0     | **95.4**   | **86.5**  | **92.1**  | **82.7**  | **88.2** | **86.3** |
>
> With only half the parameters (0.35%), MoG already matches or surpasses the performance of baselines. These results confirm that MoG achieves higher parameter efficiency while preserving the layer-specific flexibility, outperforming existing parameter sharing and MoE-based LoRA methods.

---

> ### Author Response · Authors · 2025-11-20
> **Response to Reviewer NQti, Part 2**
>
> > **Weakness 2.** The idea of ‘less is more’ in PEFT has been discussed… so the contribution may be limited.
>
> **Response:** Our contribution goes beyond simply applying a "less is more" philosophy.  MoG introduces a **structure-aware** **group-sharing mechanism** that leverages the model’s intrinsic structural features to guide parameter sharing, and uses a combination of **gating** and **lightweight per-layer fusion** to balance global and unique information.
>
> - **Structure-Aware Grouping:** MoG is inspired the observation that neighboring layers often exhibit high functional similarity. We analyzed the model’s weights and performed clustering to determine which layers share close functional behavior and which layers are distinct. This analysis allows us to design a grouping strategy that aligns parameter sharing with the model’s inherent structure. This ensures that redundancy is reduced where it truly exists, without compromising layers that require specialization.
> - **Global Knowledge Integration:** MoG uses a lightweight gating network to aggregate information from all groups, allowing each layer to adaptively extract global features, thereby achieving higher representational capacity and stronger expressive power.
> - **Efficient Layer-wise Fusion:** MoG assigns each layer only a lightweight $r×r$ fusion matrix to restore layer-specific adaptation. Compared with existing parameter-sharing methods such as BSLoRA, which allocates a full LoRA module to every layer, MoG achieves higher parameter efficiency while preserving layer-wise expressiveness.
>
>
> > **Weakness 3.** The paper lacks quantitative analysis of how much MoG actually reduces parameters… GPU memory comes mainly from the base model, so the reduction may be limited.
>
> **Response:**
>
> In Section 4.2 (Table 2) of the paper, we have already provided a detailed analysis of MoG’s parameter efficiency. The parameter formulas for LoRA and MoG are shown below:
>
> | Method | Parameter Formula     |
> | ------ | --------------------- |
> | LoRA   | $Lr(d+k)$             |
> | MoG    | $nr(d+k) + Lr^2 + Ln$ |
>
> Here, $d$ denotes the input dimension, $k$ denotes the output dimension, $L$ denotes the number of layers, $n$ denotes the number of groups, and $r$ denotes the adapter rank. Since $r \ll \min(d,k)$, $Lr^2$ and $Ln$ are negligible compared with $nr(d+k)$. Therefore, MoG achieves effective compression relative to standard LoRA.
>
> For clearer parameter comparison, we provide a detailed quantitative comparison under different configurations.
>
> |                  | llama3-8b | LoRA(r=32) | MoG(r=32,n=16) | MoG(n=8) | MoG(n=4) | MoG(n=2) | MoG(n=1) |
> | ---------------- | --------- | ---------- | -------------- | -------- | -------- | -------- | -------- |
> | Trainable params | 8030M     | 56.62M     | 28.48M         | 14.32M   | 7.24M    | 3.70M    | 1.93M    |
> | Trainable(%)     | 100       | 0.70       | 0.35           | 0.18     | 0.09     | 0.05     | 0.02     |
>
> These results demonstrate MoG’s substantial parameter reduction, achieving greater efficiency in fine-tuning LLMs. In the extreme case (n = 1), MoG reduces trainable parameters to only 1.93M, 96.6% reduction compared to LoRA (56.62M).
>
> GPU memory is primarily consumed by the base model and activation functions, but trainable parameters reduction achieved by MoG indeed decreases some memory usage, leading to practical benefits in resource-constrained settings. For example, with LLaMA2-7B, MoG reduces the adapter memory from 305 MB (LoRA) to 39 MB, an 87% reduction in the memory required for the trainable parameters compared to standard LoRA.
>
> | Base_model | **Activation** | LoRA weights | MoG weights |
> | ---------- | -------------- | ------------ | ----------- |
> | 12.85 GB   | 10.32 GB       | 305 MB       | 39 MB       |
>
>
>
> **References**
>
> [1] Song Y, Zhao J, Harris I G, et al. Sharelora: Parameter efficient and robust large language model fine-tuning via shared low-rank adaptation. arXiv:2406.10785.
>
> [2] Yuhua Zhou, Ruifeng Li, Changhai Zhou, et al. BSLoRA: Enhancing the parameter efficiency of loRA with intra-layer and inter-layer sharing. ICML2025.
>
> [3] Yao K, Gao P, Li L, et al. Layer-wise importance matters: Less memory for better performance in parameter-efficient fine-tuning of large language models. arXiv:2410.11772.
>
> [4] Zhang Y, Zhu H, Liu A, et al. Less is more: Extreme gradient boost rank-1 adaption for efficient finetuning of llms. arXiv:2410.19694.
>
> [5] Gao C, Chen K, Rao J, et al. Higher layers need more lora experts. arXiv:2402.08562.
>
> [6] Tian C, Shi Z, Guo Z, et al. Hydralora: An asymmetric lora architecture for efficient fine-tuning[J]. NIPS2024.

---

### Author Response · Authors · 2025-12-02
**Official Comment by Authors**

Dear Reviewers and ACs:

Thank you for the time and effort you have devoted to reviewing our submission. Below is a summary of the paper’s contribution and how the major reviewer concerns have been resolved.

This paper introduces MoG, a structure-aware parameter-efficient fine-tuning method. MoG leverages inter-layer functional similarity to guide adapter sharing, uses a learnable gating network to adaptively aggregate global information, and employs a lightweight fusion matrix to preserve layer-specific specialization. This design reduces LoRA’s redundancy and avoids the expressiveness degradation of hard-sharing methods. Extensive experiments further validate the method’s robustness and generality.

Across the discussion, three major concerns were raised, and all have been addressed:

**1. Motivation & Clarity of Method (Reviewers NQti & Cvwa)**

- Why grouping improves LoRA: Adjacent layers are highly similar. Sharing one adapter within each group reduces parameter redundancy while preserving the expressive capacity of distinct functional regions.
- How MoG avoids over-sharing: A lightweight gating network enables each layer to adaptively integrate global information from different groups, and a per-layer fusion matrix restores layer-specific specialization with minimal overhead.
- Role of clustering: Clustering provides a principled way to estimate the number of groups. Pre-trained and fine-tuned weights show nearly identical cluster structures, demonstrating that functional regions are stable across fine-tuning.

**2. Comparisons With Related PEFT Methods (Reviewers NQti & Cvwa)**

Reviewers noted the need for broader comparisons with parameter-sharing and MoE-based LoRA variants.

- Novelty: MoG advances prior works by combining structure-aware grouping with adaptive gating and lightweight fusion, offering greater flexibility and avoiding the performance drop in hard-sharing methods.
- Empirical validation: We added additional experiments comparing MoG with parameter-sharing and MoE-LoRA methods. MoG achieves better performance under equal and even half the parameters.
- Scalability: Additional experiments on Qwen2.5-14B further demonstrate MoG's effectiveness on larger-scale models.

This resolves concerns about breadth of comparison and novelty relative to prior works.

**3. Parameter Efficiency, FLOPs, Memory, and Application Scenarios (Reviewers NQti, Cvwa & p58V)**

We provided the following analyses:

- Parameter formulas comparing LoRA and MoG, showing up to 96.6% reduction (56.6M → 1.93M).
- Adapter memory reduction (305MB → 39MB on 7B/8B models).
- Minimal FLOPs overhead, adding only 0.04G FLOPs relative to LoRA on a 1921G baseline.
- Experiment results confirm that MoG is robust to rank and stable across learning rates.
- On multi-task benchmarks, MoG achieves the highest overall MMLU score, demonstrating its practical advantages in realistic multi-task PEFT settings.

We sincerely appreciate the reviewers’ and ACs' time and constructive feedback throughout this process. We believe these analyses and experiments comprehensively address the raised concerns. Thank you again for your consideration.

Best regards,

The Authors

---

### Meta-Review · Area_Chair_AiDC · 2025-12-21

**Summary:**

This paper proposes MoG, a group-sharing framework for parameter-efficient fine-tuning (PEFT). While the idea of grouping layers based on similarity and using adaptive gating is interesting and the paper is well-written, significant and unresolved concerns regarding the paper's contribution, experimental rigor, and positioning lead to a recommendation to reject.

The primary issue is the lack of sufficient novelty and a compelling advance over the state of the art. The core concept of balancing shared and independent parameters has been explored in prior work (e.g., DenseLoRA, VeRA, and several MoE-inspired LoRA variants), and the paper fails to situate itself clearly within or substantially beyond this existing landscape. Key baselines and relevant recent works are missing from the comparisons, making it impossible to assess the true incremental contribution.
Furthermore, the empirical validation is incomplete and raises methodological questions. The experiments are confined to single-task settings and models of up to only 8B parameters, leaving the method's scalability to larger, more realistic models and its utility in multi-task scenarios unproven. Critically, questions about the fairness of comparisons, specifically, whether all methods, including the vanilla LoRA baseline, were tuned with equal rigor (e.g., optimal learning rates), cast doubt on the reported performance gains. The static design of the fusion weights also appears disconnected from the paper's own analysis of dynamic inter-layer similarity.
Given these fundamental gaps in establishing novelty, providing comprehensive comparisons, and demonstrating robust and scalable performance, the paper does not currently meet the bar for acceptance, though the authors have provided rebuttal to address some of concerns. The authors would need to perform substantial additional work to address these concerns, which is beyond the scope of a standard revision.

**Reviewer Concerns:**

I think most of the concerns from Reviewers buem & p58V have been addressed by the rebuttal. However, some of concerns from Reviewers NQti & Cvwa are still outstanding.

**Reviewer Scores:**

Reviewers buem & p58V will most likely maintain their scores, while Reviewers NQti & Cvwa will maintain the scores.

---

### Decision · Program_Chairs · 2026-01-26

Reject